# How Few Davids Improve One Goliath: Federated Learning in Resource-Skew Edge Computing Environments

## ABSTRACT

The real-world deployment of federated learning requires orchestrating clients with widely varying compute resources, from *strong* enterprise-level devices in data centers to *weak* mobile and Web-of-Things (WoT) devices. Prior works have explored ways to downscale large models for weak devices and perform aggregation among heterogeneous models. A typical architectural assumption is that there are equally many strong and weak devices. In reality, we often see resource-skew environments where a few (1 or 2) strong devices hold a substantial amount of data resources, accompanied by a large number of weak devices. This poses challenges—the unshared portion of the large model rarely receives updates or derives benefits from the weak collaborators.

We aim to facilitate reciprocal benefits between strong and weak devices in the presence of resource skewness in federated learning. We propose RecipFL, a novel framework featuring a server-side graph hypernetwork that generates weights for personalized client models, aligning them with the unique data distributions and computational capacities of individual devices. The graph hypernetwork captures local and global structures of client models and generalizes knowledge about model weights across model architectures. Notably, RecipFL is agnostic to model scaling strategies and can enable collaboration among arbitrary neural network models. We establish the generalization bound of RecipFL through theoretical analysis and conduct extensive experiments across image classification and natural language inference tasks with various model architectures. The results show that RecipFL can improve accuracy by 3.6% and 8.7% on strong and weak devices respectively, providing incentives for strong devices to actively participate in federated learning.

## KEYWORDS
federated learning; edge computing; resource skewness

**ACM Reference Format:**
Anonymous Author(s). 2023. How Few Davids Improve One Goliath: Federated Learning in Resource-Skew Edge Computing Environments. In *Proceedings of ACM Conference (Conference'17)*. ACM, New York, NY, USA, 10 pages. https://doi.org/10.1145/nnnnnnn.nnnnnnn

## 1 INTRODUCTION

The growing demand for data privacy has catalyzed the rise of federated learning [29] as a privacy-preserving distributed learning paradigm. The real-world deployment of federated learning needs to deal with heterogeneous edge computing environments [13, 20, 33]. Typically, a few devices, often owned by large enterprises, can be very powerful to afford large models, while the vast majority are 'weak' devices that can only host small models, such as mobile and Web-of-Things (WoT) devices owned by individuals. Universally deploying homogeneous small models as required by traditional methods [17, 18] not only wastes available compute resources but also compromises performance. Ideally, we need models scaled to fit varying device capacities and perform effective model aggregation.

To support the collaboration among heterogeneous models across clients, existing approaches [5, 15, 22, 28, 30] downscale the large model for weak devices and aggregate the common parts among models. These works typically assume there are equally many strong and weak devices [15, 28, 36]. However, in reality, we often see a skewed scenario where a small number of strong devices operated by enterprises are accompanied by a large number of user-owned weak devices. For example, a smartwatch company wants to develop a human activity recognition system. The company trains a large model using a vast dataset gathered from controlled experimental environments, while their smartwatch users can participate in this process via federated learning to train small models using personal data in the wild. It is expected that small models can benefit from the large model [2, 9], however, their contribution to the large model is dubious due to their constrained capability.

In light of this gap, we address a new research question: *Can strong devices benefit from weak devices in resource-skew computing environments?* We consider an extreme scenario where the majority are weak devices and limited (1 or 2) strong devices participate in the learning, as depicted in Figure 1(a). In this scenario, the learning system heavily leans on weak devices, while the unshared portion of the large model on strong devices rarely receives updates or derives benefits from others, which brings a significant challenge to benefit strong devices from weak collaborators.

Existing approaches deploy either width-scaling that prunes the channels of the large model [5, 22, 30], or depth-scaling that performs layer-wise pruning [15, 28]. They rely on traditional weight-averaging aggregation [24, 29] to update shared layers. However, it can be destructive when the layers in small models are ill-aligned with the layers in large models. For example, if the first block of ResNet [8] is used as an independent neural network handling a completed vision recognition process, the function of its layers inevitably changes, compared to its counterpart within an entire ResNet model. When a few large ResNet models are aggregated with a majority of their smaller counterparts (i.e., ResNet with the first block), its first block may only extract shallow vision features, thus resulting in performance degradation. Even facilitated with knowledge distillation [15], there is no guarantee of benefit if knowledge is transferred from numerous small models biased by non-IID data, as corroborated by our experiment results.


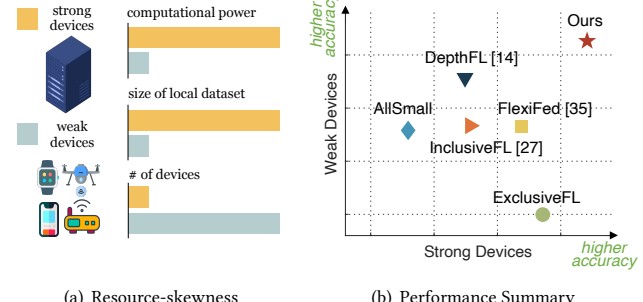

(a) Resource-skewness    (b) Performance Summary

**Figure 1: Illustration of problem setting and the performance of prior methods in resource-skew environments.**

To effectively align and aggregate models when clients employ customized model architectures, we propose RECIPFL, a novel federated learning framework that empowers the server with a graph hypernetwork tasked with generating weights for client models. Clients retain the flexibility to adapt the model architectures to their unique computational capacities, either through pruning the large model or changing architectures. The server transforms client models into directed acyclic graphs to delineate the computation flow among layers. Figure 2 presents the overview of RECIPFL. In contrast to the traditional weight-averaging aggregation method, which requires the layer to have uniform operations, sizes, and computational flows for aggregation, the graph hypernetwork can deal with arbitrary model architectures. This is achieved by encoding information about computational graphs of client models with a gated graph neural network (GatedGNN) [26] and decoding parameters with multi-layer perceptrons (MLPs). The GatedGNN captures the local and global structures of client models, sharing knowledge across model architectures even when layers vary in size or follow different computation flows. The graph hypernetwork is trained during federated learning according to the feedback (i.e., locally updated weights) from clients. The computations of hypernetwork are executed by the server and therefore do not add any additional communication or computation overhead to edge devices. We further augment weak devices by distilling knowledge from large models to smaller ones on strong devices and aggregate the knowledge by the update of the graph hypernetwork.

We theoretically analyze the generalization bound of RECIPFL and empirically evaluate RECIPFL framework across four datasets for image classification and natural language inference. We simulate non-IID client distributions and evaluate personalized client models on their own test data as real-world WoT and mobile computing applications typically require. The results show RECIPFL outperforms state-of-the-art methods across different scaling strategies and various model architectures with significant margins. Notably, RECIPFL yields improvements for both strong and weak devices, demonstrating that even devices with limited computational resources can contribute meaningfully to the learning system, thereby incentivizing strong devices to actively engage in federated learning.

Our contributions are summarized as follows:

- We address a new research question in federated learning: *Can strong devices benefit from weak devices in the presence of resource skewness?* We show the existing methods do not guarantee improvement for both types of devices.
- We propose a novel framework RECIPFL to effectively generate weights for heterogeneous client models based on graph hypernetwork, compatible with arbitrary model scaling strategies.
- We establish the generalization bound of RECIPFL through theoretical analysis and validate its performance through extensive experiments. RECIPFL outperform various state-of-the-art methods with significant margins and demonstrate that weak devices can also contribute effectively to the learning of strong devices.

**Reproducibility**. We will release the code on Github[1].

## 2 PRELIMINARIES

### 2.1 Problem Definition

We aim to build a federated learning system with $M$ clients that allows the clients to have customized model architectures $\{\mathcal{G}_m | m \in [M]\}$ that fit their specific running capabilities. Within the $M$ clients, there are a few (e.g., 1 or 2) strong devices that have enough running capacity to hold large models and the rest are weak devices having limited computing power. Denote the training set on client $m$ as $\mathbf{D}_m = \left\{ \left( x_i^{(m)}, y_i^{(m)} \right) \right\}_{i=1}^{N_m}$, where $x_i^{(m)}$ is the input data and $y_i^{(m)}$ is the label, and the data distribution of client $m$ as $\mathcal{P}_m$. Denote $\ell$ as the loss function. The goal is to learn a personalized model $f_m(\cdot; \theta_m)$ for every client $m$ that works on its own data distribution:

$$\theta^* = \arg\min_{\theta} \frac{1}{M} \sum_{m=1}^{M} \mathbb{E}_{(x,y) \sim \mathcal{P}_m} \ell(f_m(x; \theta_m), y), \quad (1)$$

where $\theta$ is the set of client model weights: $\theta = \{\theta_1, \ldots, \theta_M\}$.

### 2.2 Resource-Skew Computing Environments

Existing works typically assume the strong and weak devices are equally distributed. Interestingly, a tangential sensitivity evaluation in a prior work shows that when the strong devices are the minority, the convergence of the large models in the federated system is slow and the performance is worse than exclusively training large models without the participation of small models [28]. However, no analyses or solutions were proposed for the resource skewness problem. To have a clearer understanding of how existing methods perform in the resource-skew computing environment, we summarize the results from Section 5 by averaging the accuracies on strong and weak devices respectively across all datasets. In our experiments, we set the weak devices as the majority (e.g., 5, 20, 50, 100) and use only 1 or 2 strong device(s). For each dataset, we allocate 50% of the entire dataset to the strong device(s), with the remaining data assigned to small devices following Dirichlet distributions. We craft two naive baselines: AllSmall, which trains small models on all devices via federated learning, and ExclusiveFL, which lets strong devices run large models and weak devices run small models and conducts federated learning within their respective groups. Note that when only 1 strong device exists, it conducts centralized learning under ExclusiveFL. Additionally, we include comparisons with existing federated methods for heterogeneous models [15, 28, 36]. For each method, we evaluate client models on their own test data

[1]https://github.com/anonymous

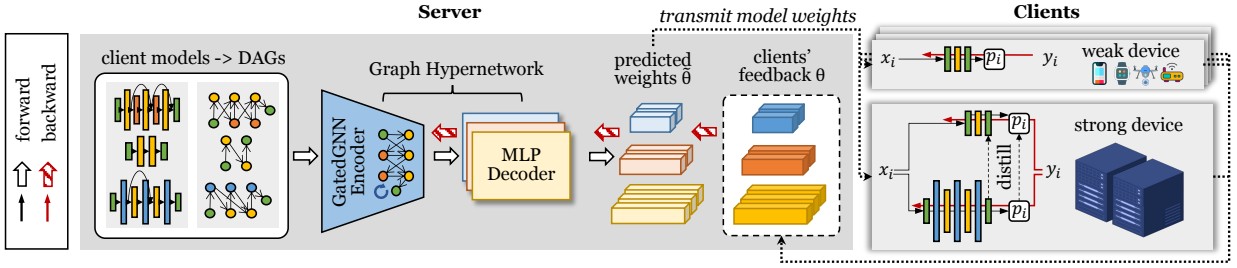

**Figure 2: Overview of RECIPFL framework. The server transforms client models into directed acyclic graphs (DAGs) to represent the computation flow among operations and trains a graph hypernetwork to generate weights for customized client models.**

sampled from clients' data distributions. The summary is presented in Figure 1(b), from which we draw the following observations:

- *Small models are insufficient for strong devices:* By comparing AllSmall and ExclusiveFL on strong devices, we see that training small models with collaboration from weak devices yields lower accuracy compared to exclusively training large models.
- *Weak devices benefit from collaboration with strong devices:* By comparing existing methods and AllSmall on weak devices, we see the existing methods generally show a higher accuracy.
- *Strong devices derive minimal benefits from weak devices with existing methods:* For strong devices, we see the accuracy of existing methods is always lower than ExclusiveFL.
- *Existing methods could enhance the performance of one type of model but struggle to improve both.* For example, DepthFL achieves high accuracy on weak devices but does not perform well on strong devices. FlexiFed achieves higher accuracy on strong devices than DepthFL but shows less improvement on weak devices.

Recognizing the limitations of prior methods, we aim to facilitate reciprocal benefits between strong and weak devices in resource-skew computing environments.

## 3 OUR RECIPFL FRAMEWORK

### 3.1 Federated Training

The pseudo code of the training process is presented in Algorithm 1. During each round of training, the server initiates the process by randomly selecting a subset of clients, denoted as $S_t$, to conduct local updates. The server utilizes the graph hypernetwork to produce model weights $\{\tilde{\theta}_m | m \in S_t\}$, sends the weights to selected clients and waits for their feedback. At the client side, the client performs local updates by training the client model $f_m$ with its local dataset $D_m$. The training objective at client $m$ is to minimize the loss:

$$\underset{\theta}{\text{argmin}}\ \mathcal{L}_m(\theta_m) = \underset{\theta}{\text{argmin}}\ \frac{1}{N_m} \sum_{i=1}^{N_m} \ell\left(f_m\left(x_i^{(m)}; \theta_m\right), y_i^{(m)}\right), \quad (2)$$

where $N_m$ is the number of samples in the local dataset $D_m$. Let $\eta_c$ be the learning rate for local training at the client. Starting with the initial value $\theta_m = \tilde{\theta}_m$, the client updates $\theta_m$ as follows:

$$\theta_m \leftarrow \theta_m - \eta_c \nabla_{\theta_m} \mathcal{L}_m(\theta_m). \quad (3)$$

After local training, the clients send the updated model weights back to the server. The server then calculates the change in local

---

**Algorithm 1: RECIPFL Framework**

**Input** : Communication rounds $T$, number of selected clients per round $|S_t|$, local training epochs $E$, client descriptors $\{a_m | m \in [M]\}$ and model architectures $\{\mathcal{G}_m | m \in [M]\}$.

**Output**: A graph hypernetwork that generates personalized model weights for heterogeneous client models.

**Server executes:**

**for** $t = 1, \ldots, T$ **do**
  Select a subset $S_t$ of clients at random;
  **for** $m \in S_t$ **do**
    $\tilde{\theta}_m \leftarrow \text{GHN}(\mathcal{G}_m, a_m; \phi)$;
    $\theta_m \leftarrow \textbf{ClientUpdate}(m, \tilde{\theta}_m)$;
    $\Delta\theta_m \leftarrow \theta_m - \tilde{\theta}_m$;
  Update GHN: $\phi \leftarrow \phi - \eta_s \sum_{m \in S_t} (\nabla_\phi \theta_m)^T \Delta\theta_m$;

**return** $\text{GHN}(\cdot; \phi)$;

**ClientUpdate**$(m, \tilde{\theta}_m)$:

$\theta_m \leftarrow \tilde{\theta}_m$;
**for** $e = 1, \ldots, E$ **do**
  Partition $D_m$ into mini-batches $\bigcup_{i=1}^{j_m} B_i^{(m)}$;
  **for** $i = 1, \ldots, j_m$ **do**
    $\theta_m \leftarrow \theta_m - \eta_c \nabla_{\theta_m} \mathcal{L}_m(\theta_m; B_i^{(m)})$ ;

**return** $\theta_m$ to server ;

---

model parameters $\Delta\theta_m = \theta_m - \tilde{\theta}_m$, and uses the chain rule $\nabla_\phi \mathcal{L}_m = (\nabla_\phi \theta_m)^T \nabla_{\theta_m} \mathcal{L}_m$ to update the graph hypernetwork parameter $\phi$:

$$\phi \leftarrow \phi - \eta_s \sum_{m \in S_t} (\nabla_\phi \theta_m)^T \Delta\theta_m, \quad (4)$$

where $\eta_s$ is the learning rate for updating the graph hypernetwork. By adopting graph hypernetwork, we modify Equation 1 and formulate the new learning objective as:

$$\underset{\phi}{\text{argmin}}\ \hat{\mathcal{L}}(\phi, D). \quad (5)$$

where $\hat{\mathcal{L}}(\phi, D) = \frac{1}{M} \sum_{m=1}^{M} \mathcal{L}_m(\text{GHN}(\mathcal{G}_m, a_m; \phi))$ is the average empirical loss on dataset $D = \{D_m\}_{m=1}^{M}$.

Importantly, the graph hypernetwork resides on the server, with all computations related to the graph hypernetwork executed solely

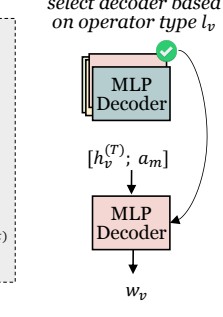

(a) Encoding        (b) Decoding

**Figure 3: Graph hypernetwork architecture.**

by the server. This design ensures it does not impose any additional communication or computational overhead on clients.

## 3.2 Weight Generation with Graph Hypernetwork

**Representation of target network architectures**. We represent the computational graph of a neural network model as a directed acyclic graph, denoted as $\mathcal{G}(\mathcal{V}, \mathcal{E})$, where the nodes $\mathcal{V}$ are the operators (e.g., convolution, pooling, linear layer, etc.) and the directed edges $\mathcal{E}$ describe the computation flow in the order of forward propagation among the operators. Conventional graph hypernetworks are inefficient to deal with repeated similar local connection patterns in deep networks such as the ResNet blocks in ResNet-152. To enhance the ability to distinguish local connection patterns in target networks, we inform the graph hypernetwork about the model parameters of the target network at the current training round. To do so, the node features $\{h_v | v \in C\} = \{[l_v, q_v] | v \in C\}$ consist of two parts: (1) one-hot vectors $l_v$ indicating the operations performed by the node, and (2) the current parameters $q_v$ of the operators. A linear embedding layer transforms the one-hot vector $l_v$ to a dense vector and a Transformer encoder [34] maps the variable-length node parameters $q_v$ to a fixed-dimensional vector. The linear embedding layer and the Transformer encoder are learnable and are updated during training.

**Graph Hypernetwork Architecture**. As depicted in Figure 3, the graph hypernetwork consists of an encoding process that extracts features from node information and a decoding process that predicts weights for parametric operators according to the encoded features.

During the encoding phase, a Gated Graph Sequence Neural Network (GatedGNN) [26] is employed to conduct $\tau$ steps of graph propagation within $\mathcal{G}(\mathcal{V}, \mathcal{E})$ of the target network. During the propagation, the GatedGNN topologically traverses the nodes in both forward and backward directions, iteratively conducting message passing and updating node features. For the $t$-th propagation step, the GatedGNN first forward traverses nodes. Every node $v$ receives messages from its incoming nodes and sends messages to its outgoing nodes. Denote the incoming nodes to node $v$ as $\mathbf{IN}(v)$. The message function is modeled with an MLP shared among all the nodes. The message received by node $v$ at step $t$ is:

$$m_v^{(t)} = \sum_{u \in \mathbf{IN}(v)} \text{MLP}(h_u^{(t)}) \tag{6}$$

The node feature vector $h_v^{(t)}$ is then updated based on the aggregated message $m_v^{(t)}$ and the feature vector of node $v$ at step $t-1$ using a Gated Recurrent Unit (GRU) cell:

$$h_v^{(t)} = \text{GRU}(h_v^{(t-1)}, m_v^{(t)}) \tag{7}$$

After traversing $\mathcal{G}(\mathcal{V}, \mathcal{E})$ in forward propagation, the GatedGNN reverses the traversal direction and updates the node features again, i.e. receives messages from its incoming nodes along backward passes and sends to its outgoing nodes.

During the decoding phase, we use an individual MLP as the decoder for each type of parametric operator to generate parameters. To further support personalization, we introduce client descriptors $\{a_m | m \in [M]\}$ that describe the data characteristics of every client $m$. This descriptor is provided as input to the MLP decoder. Specifically, we use the class distribution of local training samples as the client descriptor. Alternatively, the client descriptor can simply be the client IDs, and in that case, a linear embedding layer can be used to transform them into client embeddings, enabling the learning of the client embedding through training. Let $\text{MLP}_l(\cdot)$ represent the decoder for the $l$-type operator. $\text{MLP}_l$ operates on the concatenation of the node embedding and the client embedding, denoted as $[h_v^{(\tau)}, a_m]$, and generates the parameters for the node. The resulting set of generated weights for the target network is:

$$w = \{w_v | v \in \mathcal{V}\} = \{\text{MLP}_{l_v}([h_v^{(\tau)}, a_m]) | v \in \mathcal{V}\} \tag{8}$$

To handle different dimensionalities of layers within the same operator type, the outputs of the decoder are reshaped through tiling and concatenation to match the shape of the target layers following common practices in graph hypernetworks [16, 38].

## 3.3 Knowledge Transfer from Strong to Weak Devices

To further enhance the learning of small models, we leverage the computing resources on strong devices and employ regularizations to distill knowledge from large models to small ones.

For strong devices, we let the central graph hypernetwork generate weights for both small and large models. Denote the small and large model at the strong device $m$ as $f_m^S$ and $f_m^L$ respectively and the corresponding model parameters are $\theta_m^S$ and $\theta_m^L$. After training the large model $f_m^L$, we proceed to train the small model and distill knowledge from the large model. We introduce an additional cross-entropy loss term $CE(\cdot)$ to let the small model mimic the prediction probabilities of the large model. In addition, we add a KL-divergence loss term $D_{KL}(\cdot)$ to align the feature spaces between the small and large models. Denote the softmax probability distributions of features generated by the last hidden layer before the classifier of the small model and the large model as $p_i^S$ and $p_i^L$ respectively. The training objective for the small model $f_m^S$ on large device $m$ is to minimize the following loss:

$$\begin{aligned} \mathcal{L}_m^S(\theta) = \frac{1}{n} \sum_{i=1}^n [&CE(f_m^S(x_i; \theta_m^S), y_i) \\ &+ CE(f_m^S(x_i; \theta_m^S), f_m^L(x_i; \theta_m^L)) + D_{KL}(p_i^S \| p_i^L)]. \end{aligned} \tag{9}$$

After local training, the updated weights of both small and large models are transmitted to the server and used in conjunction with weights from other selected clients for the update of the graph

hypernetwork. This knowledge transfer mechanism helps small models benefit from insights learned by strong devices.

## 4  ANALYSIS ON GENERALIZATION BOUND

In this section, we analyze RecipFL method theoretically and establish a generalization bound.

Consider a training set on clients $\mathbf{D}_m = \left\{ \left( x_i^{(m)}, y_i^{(m)} \right) \right\}_{i=1}^N$ for some natural number $N \geq 1$, i.e. we sample uniformly $N$ training data from each data distribution $\mathcal{P}_m$ on client $m$ for $m = 1, \cdots, M$.

Assume the loss function $\ell$ takes value in $[0, 1]$, or equivalently with rescaling, $\ell$ is bounded. Let $d$ be the dimension of the graph hypernetwork parameter $\phi$ and assume $\phi \in [-R, R]^d$ for some large $R > 0$. Finally, assume the loss $l$ is Lipschitz with respect to $\phi$ with Lipschitz constant $K > 0$, i.e. $|\ell\left( f_m\left( x; \text{GHN}(\mathcal{G}_m, a_m; \phi) \right), y \right) - \ell\left( f_m\left( x; \text{GHN}(\mathcal{G}_m, a_m; \phi') \right), y \right)| \leq K\|\phi - \phi'\|$ for all $x, y$ and $m = 1, \ldots, M$. Here $\|\cdot\|$ denotes the Euclidean distance on $\mathbb{R}^d$. Define the expected loss as:

$$\mathcal{L}(\phi) = \frac{1}{M} \sum_{m=1}^M \mathbb{E}_{(x,y) \sim \mathcal{P}_m} \ell(f_m(x; (\mathcal{G}_m, a_m; \phi)), y). \quad (10)$$

THEOREM 4.1. *If the number of samples of each client's data distribution satisfies*

$$N \geq \max\left\{ \frac{4d}{M\epsilon^2} \log\left\lceil \frac{4RK\sqrt{d}}{\epsilon} \right\rceil + \frac{4}{M\epsilon^2} \log\frac{4}{\delta}, \frac{1}{\epsilon^2} \right\}, \quad (11)$$

*then with probability at least $1 - \delta$ with respect to the probability distribution on $\mathbf{D} = \{\mathbf{D}_m\}_{m=1}^M$, $\mathcal{L}(\phi) < \hat{\mathcal{L}}(\phi, \mathbf{D}) + \epsilon$ for every $\phi$.*

The proof and more details are given in the Appendix. From Equation 11, we observe that the number of training samples $N$ per device required for generalization is negatively related to the number of devices $M$, which suggests that introducing new weak devices to the system can help lower the threshold for generalization. Moreover, when there is a strong device that possesses a large amount of data, it can also lower the threshold for weak devices. For example, if there is one strong device and $M$ weak devices, we can regard the strong one as $k$ virtual devices, which increases the total number of devices to $M + k$, and thereby lower the threshold for the number of samples on weak devices. The only requirement is that the strong device then needs to take on $k$ times more data samples than that is required for a weak device. In Section 5.4, we show how the data allocation on strong and weak devices affects the performance through exploratory studies. The results of the experiment align with our theoretical findings.

## 5  EXPERIMENTS

### 5.1  Experiment Setup

We summarize the configurations of the experiments in Table 2. Details are presented as follows.

**Datasets**. We evaluate RecipFL on the image classification task with CIFAR-10 [19], CIFAR-100 [19], MNIST [21], and the natural language inference task with MNLI [37]. We simulate quantity skew where strong devices possess a dominant amount of data, as it often occurs in realistic resource skew environments. We allocate 50% of the entire dataset to the large devices, while the small devices

**Table 1: RecipFL is compatible with various ways of model scaling, showing more flexibility than existing solutions.**

| Scaling Strategy | HeteroFL | InclusiveFL | FlexiFed | DepthFL | RecipFL (ours) |
|---|---|---|---|---|---|
| depth-wise | | ✓ | ✓ | ✓ | ✓ |
| width-wise | ✓ | | | | ✓ |
| architecture-wise | | | | | ✓ |

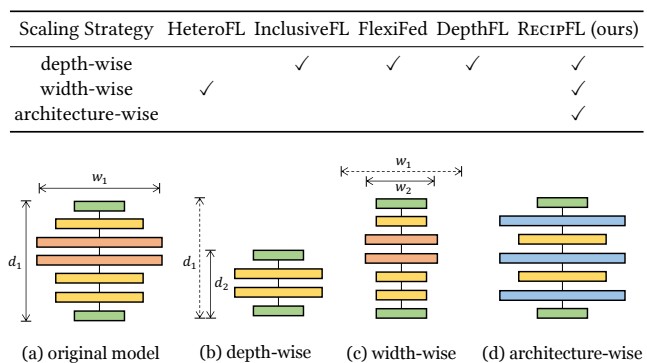

(a) original model    (b) depth-wise    (c) width-wise    (d) architecture-wise

**Figure 4: Illustration of model scaling strategies. The rectangle blocks represent the layers in neural networks. Different colors indicate different operations (e.g., convolution).**

evenly share the remaining half. This ensures the total amount of the data owned by weak devices is comparable to that owned by strong devices, making it possible for weak devices to contribute to the model enhancement of strong devices. Note that we conduct exploration studies in Section 5.4 to investigate the impact of data ratio by changing this configuration. To simulate non-IID client distributions, we follow the prior work [11, 15] and employ Dirichlet distribution $Dir(\alpha = 0.5)$ to sample the class distribution for every client. For the large device, we assume it follows the universal distribution due to its substantial data volume.

**Model architectures**. To evaluate the robustness of our framework, we experiment with various model architectures. The large models include ResNet-18 [8], DenseNet-121 [12], LeNet-5 [21] and BERT [4]. Our framework is compatible with different ways of model scaling and we test all three scaling strategies shown in Figure 4. For depth-scaling, we follow [15] and regard the first block of ResNet-18 and DenseNet-121 as the small models. For width-scaling, we follow [5] and shrink the channels and hidden layers of the large model based on a scaling ratio. In order to achieve comparable model sizes with depth-scaling, we carefully set the scaling ratio for width-scaling by comparing the parameters in the depth-scaled models to those in the large models. In addition, we craft a smaller version of LeNet-5 which reserves the first block of LeNet-5 and scales the rest layers along the width. By doing this, we enable both depth- and width-wise aggregation for the comparison of existing methods. For architecture-wise scaling, We use DistilBERT [31] as the smaller version of BERT.

**Enabling Fine-tuning from Pretrained Models**. RecipFL can support fine-tuning by inserting adapters [10], a small set of new parameters, and classification heads into pretrained models. During training, only the adapters and classification heads are updated and communicated between the server and clients, while the other layers are fixed at local. We initialize BERT[2] and DistilBERT[3] with pretrained weights provided by HuggingFace.

---

[2]https://huggingface.co/bert-base-uncased
[3]https://huggingface.co/distilbert-base-uncased

Table 2: Federated Learning Configurations.

| Dataset | # of devices | | Data allocation (%) | | Large model | # of parameters | | | Pretrained? |
|---|---|---|---|---|---|---|---|---|---|
| | Strong | Weak | Strong | Weak | | Original | Depth-scaled | Width-scaled | |
| CIFAR-10 | 1 | 5 | 50% | 10% | ResNet-18 | 11M | 450K | 444K | ✗ |
| CIFAR-100 | 1 | 50 | 50% | 1% | DenseNet-121 | 1M | 258K | 276K | ✗ |
| MNIST | 2 | 100 | 25% | 0.5% | LeNet-5 | 44K | 5.6K (scaled in depth & width) | | ✗ |
| MNLI | 1 | 20 | 50% | 2.5% | BERT | 110M | 67M (DistilBERT) | | ✓ |

Table 3: Experiment results (average accuracy and standard deviation). RᴇᴄɪᴘFL consistently outperforms the compared methods across all datasets and model scaling strategies, benefiting both strong and weak devices.

| Scaling | Method | CIFAR-10 | | CIFAR-100 | |
|---|---|---|---|---|---|
| | | Strong | Weak | Strong | Weak |
| Depth | AllSmall | 64.34±2.14 | 68.50±3.42 | 17.86±2.56 | 25.56±3.11 |
| | ExclusiveFL | 84.85±1.85 | 59.11±4.22 | 32.21±3.81 | 19.22±2.07 |
| | FlexiFed [36] | 82.86±1.77 | 67.66±3.93 | 28.60±3.48 | 27.84±2.98 |
| | InclusiveFL [28] | 83.22±0.47 | 67.66±3.14 | 18.98±3.49 | 28.71±2.87 |
| | DepthFL [15] | 73.90±1.49 | 78.16±1.48 | 22.08±3.58 | 36.83±2.87 |
| | **RᴇᴄɪᴘFL** | **85.28±0.22** | **78.65±1.35** | **41.63±2.24** | **45.52±3.12** |
| Width | AllSmall | 82.86±1.77 | 78.90±2.87 | 29.80±3.32 | 37.90±2.83 |
| | ExclusiveFL | 83.96±1.97 | 70.65±3.99 | 32.22±6.66 | 24.49±3.52 |
| | HeteroFL [5] | 84.76±1.19 | 77.93±2.92 | 26.51±2.70 | 39.05±2.82 |
| | **RᴇᴄɪᴘFL** | **85.06±0.13** | **82.88±1.29** | **43.64±2.84** | **42.00±3.88** |

| Method | MNIST | | MNLI | |
|---|---|---|---|---|
| | Strong | Weak | Strong | Weak |
| AllSmall | 94.84±1.34 | 92.01±3.53 | 73.47±0.52 | 82.13±2.89 |
| ExclusiveFL | 96.21±0.59 | 83.59±3.58 | 80.20±0.20 | 70.52±6.04 |
| FlexiFed [36] | 96.13±1.04 | 90.15±4.78 | 79.65±0.18 | 82.31±6.15 |
| InclusiveFL [28] | 91.18±1.29 | 87.42±3.35 | 79.87±0.30 | 81.17±4.31 |
| DepthFL [15] | 96.92±0.68 | 92.94±2.72 | 77.11±0.90 | 80.92±6.64 |
| HeteroFL [5] | 83.47±1.96 | 79.17±6.71 | 79.65±0.18 | 82.31±6.15 |
| **RᴇᴄɪᴘFL** | **97.99±0.58** | **95.44±2.28** | **82.78±0.57** | **83.37±4.72** |

**Compared methods**. First, we construct two naive baselines based on the classical federated learning algorithm FedAvg [29]:

- **AllSmall**: All clients deploy the small models to compromise the smallest running capacity and conduct federated learning.
- **ExclusiveFL**: Clients with the same level of capacity are equipped with the same model, i.e., strong devices deploy large models while weak devices deploy small models. Each type of device performs federated learning exclusively. When there is only 1 strong device, it conducts centralized learning.

The performance of weak devices under AllSmall and that of strong devices under ExclusiveFL serve as reference points for assessing whether a method enhances the performance of weak or strong devices. We then compare RᴇᴄɪᴘFL with state-of-the-art methods for federated learning with heterogeneous models:

- **HeteroFL** [5] adopts width-scaling. The channels and hidden layers are scaled according to a fixed ratio. The global layer updates a subset of weight parameters correspondingly from scaled layers and updates all the parameters from unscaled layers by weight averaging.
- **FlexiFed** [36] identifies common base layers across client models and clusters personal layers into groups. The same group of personal layers have identical operations and sizes. Then, it fuses the knowledge contained in common base layers and clustered personal layers by weight averaging.
- **InclusiveFL** [28] adopts depth scaling. The common base layers are aggregated via weight averaging. It also distills knowledge from the classifier of the large model to its shallow counterpart by calculating a gradient momentum as the average over updates

of the deep layers (pruned in the small model) in the large model and injecting it to the last encoding layer in the small model.

- **DepthFL** [15] scales the large model along the depth and creates local models with different depths and classifiers. The shared layers are aggregated (i.e., averaged) across the clients. It is further equipped with a self-distillation strategy to transfer knowledge among deep and shallow classifiers if available at local.

Table 1 showcases the downscaling strategies that prior methods work on. For architecture-wise scaling settings, these methods identify shared layers (e.g. classification heads) for aggregation. Note that HeteroFL can be generalized to FlexiFed for the common base layers reserved in depth-scaling settings (i.e., the scaling ratio of those layers is 1).

**Federated learning configuration**. We evaluate each client model on its respective test data which is sampled from the client's data distribution. To obtain personalized models, for every compared method, we fine-tune the client models on their local training dataset for one round after receiving the model weights from the server. During training, the server randomly samples $|S_t| = \min(M, 10)$ clients per communication round. We set the number of communication rounds $T$ according to the convergence speed of each task. Specifically, we set $T = 50$ rounds for CIFAR-10 and MNLI, $T = 100$ rounds for MNIST, and $T = 500$ rounds for CIFAR-100. Since RᴇᴄɪᴘFL trains both small and large models on strong devices to enable knowledge transfer, to ensure a fair comparison, we also train both types of models on strong devices for the compared methods (i.e., FlexiFed, HeteroFL, InclusiveFL, and DepthFL). During evaluation, only the target client model is evaluated for

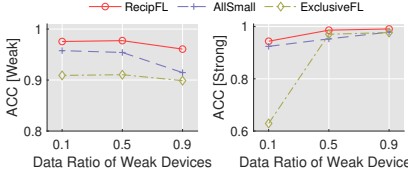 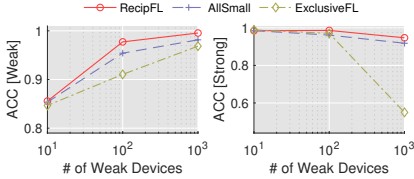 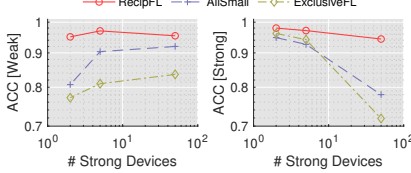

(a) Impact of data ratio

(b) Scalability: Varying weak devices

(c) Degree of skewness: Varying strong devices

**Figure 5: Exploratory Studies. RᴇᴄɪᴘFL exhibits superior scalability and robustness across a range of resource skewness scenarios compared to the baselines, consistently enhancing the performance of both strong and weak devices.**

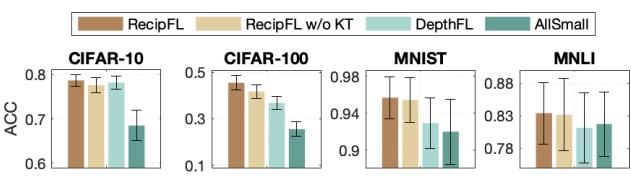

**Figure 6: Ablation study: Performance of weak devices.**

every client. The experiments are repeated 5 times. Following [25], we report the accuracy of the last 20% rounds averaged over strong and weak devices respectively, along with standard deviations.

## 5.2 Main Experiment Results

The experiment results are presented in Table 3. Note that the average accuracy on weak devices may appear higher than that on strong devices since the evaluation is based on every client's data distribution and the weak devices may only have a small subset of classes, making it easier to get higher accuracy. By comparing the performance of the state-of-the-art methods (HeteroFL, FlexiFed, InclusiveFL, and DepthFL) with the two naive baselines (AllSmall and ExclusiveFL), we see that existing methods can often achieve equal or superior performance compared to training small models on all devices (AllSmall). However, they do not perform as well as ExclusiveFL on strong devices. Furthermore, there is no scaling strategy that consistently outperforms others. For example, width-scaling works better than depth-scaling on CIFAR-10 and CIFAR-100 with the ResNet and DenseNet architectures but it (i.e., the result of HeteroFL) lags behind depth-scaling on MNIST with LeNet.

RᴇᴄɪᴘFL consistently outperforms the compared methods across all datasets, regardless of the model scaling strategies, demonstrating its capability to generalize knowledge across different model architectures. Notably, RᴇᴄɪᴘFL shows its ability to improve the model performance on both strong devices and weak devices. Moreover, when fine-tuning from the pretrained weighted of BERT and DistilBERT, RᴇᴄɪᴘFL also shows a better performance compared to the baselines. More details about the performance with respect to the communication round are presented in Figure 7 in the Appendix. In general, RᴇᴄɪᴘFL achieves a better performance and is more stable than the compared methods.

## 5.3 Ablation Study

In Section 3.3, we introduced the knowledge transfer mechanism on the base of our graph hypernetwork design to further enhance

the performance of small models. To evaluate the effectiveness of this design, we craft an ablated version of RᴇᴄɪᴘFL without knowledge transfer, denoted as **RᴇᴄɪᴘFL w/o KT**. We evaluate the model performance on weak devices and compare the performance of RᴇᴄɪᴘFL, RᴇᴄɪᴘFL w/o KT, AllSmall, and DepthFL (the best-performing baseline in depth scaling setting). The results are presented in Figure 6. We observe that RᴇᴄɪᴘFL w/o KT already exhibits significant improvements over the naive baseline AllSmall across all datasets, and it can often outperform the state-of-the-art method DepthFL. However, the comparison between RᴇᴄɪᴘFL and RᴇᴄɪᴘFL w/o KT indicates that the inclusion of knowledge transfer leads to even better small models. The knowledge (i.e., prediction logits and feature distribution) from strong devices contribute to the improvement.

## 5.4 Exploratory Studies

To gain deeper insights into the behavior of RᴇᴄɪᴘFL across diverse resource-skew conditions, we conduct exploratory studies using the MNIST dataset with the LeNet architecture and involve comparisons between RᴇᴄɪᴘFL, AllSmall, and ExclusveFL.

**Impact of data ratio between two types of devices**. We aim to understand how much data on weak devices is required in order to provide improvement for strong devices. We vary the ratio of total data owned by each type of device to the entire dataset among {0.1, 0.5, 0.9}. The number of devices remains the same as in the main experiment, i.e., 100 weak devices and 2 strong devices. The results are presented in Figure 5(a). On weak devices, we observe a decrease in performance as the total data ratio increases. This can be attributed to the reduced availability of data samples for training the large models, resulting in decreased large model performance, and consequently, the knowledge transferred to the small models becomes less effective in enhancing their performance. This observation also aligns with the implication derived from the theoretical analysis on the generalization bound: When the number of devices $M$ remains unchanged, weak devices need more training samples to achieve generalization if the data quantity on the large devices decreases. On strong devices, we observe that as the data ratio increases, the large model benefits from a larger quantity of data for training. Remarkably, when the data ratio allocated to the large model is small (less than 50%), the performance gap between RᴇᴄɪᴘFL and ExclusiveFL becomes more pronounced. This observation suggests that when the total data owned by weak devices is comparable to that owned by strong devices, the weak devices are more likely to contribute significantly to the model improvement of the strong devices. Based on these findings, we recommend that the federated

system involves a substantial number of weak devices, as this can lead to substantial improvements for strong devices.

**Scalability and degree of skewness**. To evaluate the scalability of the federated systems, First, we vary the number of weak devices among {10, 100, 1000}. The number of strong devices and the data ratio are kept the same as in the main experiments where the two large devices own 50% of the whole dataset and the weak devices share the rest. The results are shown in Figure 5(b). When increasing the number of weak devices, the strong devices get selected for local updates less frequently. Consequently, within the same communication rounds, the performance of ExclusiveFL and AllSmall on strong devices degrades. Conversely, the overall sampling rate of weak devices becomes higher as the number of weak devices increases, leading to improved performance of weak devices. In contrast, RecipFL demonstrates an impressive capability to memorize client model weights and generalize them across different architectures effectively. As a result, the clients can achieve good performance even with a reduced client sampling ratio. This suggests RecipFL is more scalable than the baselines. Then, we increase the number of strong devices from 2 to 5 and 50 while keeping the number of weak devices constant at 100. The strong devices equally share 50% of the entire dataset. Figure 5(c) presents the results. We see that RecipFL always achieve better performance than the two baselines. These results highlight the robustness of RecipFL across varying degrees of skewness, whether in a relatively balanced environment (with a strong-to-weak device ratio of 1:2) or an extremely skewed one (with a strong-to-weak device ratio of 1:500).

## 6 RELATED WORK

**Federated Learning with Heterogeneous Models**. Federated learning [29] has emerged as a privacy-preserving approach to collaborative machine learning across decentralized devices. Traditional federated learning methods [14, 17, 18, 24, 40] have primarily focused on homogeneous model architectures, where all participating devices train identical models. These methods often fall short in addressing the inherent system heterogeneity found in real-world edge computing environments. As such, the exploration of federated learning in the context of diverse computational capacities has given rise to novel approaches that facilitate collaboration among heterogeneous models [1, 6, 23, 39]. Existing work explores two directions to facilitate the collaboration among devices with different running capacities: (1) how to scale the large model for weak devices and (2) how to effectively aggregate the models with different sizes. In the first direction, methods are proposed to prune the model along the model depth [15, 28] and layer width [5, 22, 30]. As the first work to deal with the model heterogeneity problem in federated learning, HeteroFL [5] proposes to scale the width of hidden channels in the large model to reduce the computation complexity. More recently, DepthFL [15] proposes to prune the deepest layers in the large model to work as small client models. It reports better performance than width scaling. In the second direction, typical practices [36] are to identify shared patterns (i.e., layers) in local models and aggregate the common parts. Recent methods like InclusiveFL [28] and DepthFL [15] take one step further which leverage knowledge distillation for transferring knowledge among deeper layers and shallow layers to enhance the performance of

the small models. These approaches have shown promise in accommodating device-specific requirements and resource constraints. However, the reliance on a particular scaling strategy and the naive weight averaging-based aggregation constrain model performance in the presence of resource-skewness. In this work, we design a more effective way to generalize knowledge about model weights across different models by training a graph hypernetwork.

**Hypernetworks**. A hypernetwork [7] is a neural network that predicts the model parameters of another neural network (i.e., the target network). Hypernetworks have demonstrated the potential in meta-learning scenarios [35], facilitating fast adaptation to new tasks, as they capture the common knowledge among tasks via the weight generation mechanism. Prior work [32] has explored its use in federated learning by training a hypernetwork at the server to generate personalized model weights while preserving the effective parameter-sharing feature of hypernetworks. This previous work uses a linear-structured hypernetwork that only works with homogeneous model architectures. Graph hypernetwork [16, 38] was originally proposed for neural architecture search as it can effectively encode the computational graph information of various neural networks. There has been an initial try on leveraging graph hypernetworks for generating weights across different client models [27]. However, the prior work trains local hypernetworks at clients and aggregates them by weight averaging at the server following a typical federated training process which requires high computational budgets at clients and is impractical for resource-constrained small devices. In contrast, RecipFL equips the graph hypernetwork at the server and we design ways to update the graph hypernetwork based on predicted weights and clients' feedback. The computations of hypernetwork are executed only by the server and therefore do not add any additional communication or computation overhead to the edge devices.

## 7 CONCLUSIONS AND FUTURE WORK

We study the problem of federated learning in the presence of resource skewness among devices, specifically, when the majority are weak devices and there are only limited (1 or 2) strong devices. We show that existing methods do not guarantee performance improvement for both types of devices. We propose RecipFL framework that trains a central graph hypernetwork to enable the collaboration of clients with heterogeneous model architectures in order to fit specific running capacities. RecipFL is agnostic to model scaling strategies and is able to generalize knowledge about model weights across different neural network architectures. Our experiment results show that RecipFL can outperform state-of-the-art methods with significant margins and demonstrate that even weak devices can contribute effectively to the learning system, providing strong devices with an incentive to participate. In future work, we plan to design mechanisms to adaptively adjust the model size in response to the dynamic changes in the running capacity of devices caused by user usage. This will enable efficient utilization of compute resources during learning. Together with our proposed framework, we envision our solutions will create a viable, more powerful, and useable alternative to current large model services, alleviating privacy and efficiency concerns by facilitating edge-based learning without the need to transmit user input to central servers.

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

# A APPENDIX

## A.1 More Experiment Results

**Performance w.r.t. Communication Rounds**. Figure 7 shows the performance of all compared methods with respect to the communication round. We observe that RECIPFL often achieve higher accuracy in fewer rounds compared to the baseline methods.

## A.2 Theoretical Derivations

**Notations**. We will give the proof of Theorem 4.1 using the results of Baxter [3]. Let us introduce the notations and definitions before we state a key theorem from Baxter (Theorem 18 and Corollary 19) from which the main results of the paper are derived.

Let $\mathcal{X}$ be the input space and $\mathcal{Y}$ the output space. Let $\mathcal{P}_1, \ldots, \mathcal{P}_M$ be $M$ probability measures on $\mathcal{X} \times \mathcal{Y}$. For every $m = 1, \ldots, M$, sample $(x^{(m)}, y^{(m)})$ from the distribution $\mathcal{P}_m$, and abbreviate $\mathcal{L}_m(\phi) = \ell\left(f_m\left(x^{(m)}; \text{GHN}(\mathcal{G}_m, a_m; \phi)\right), y^{(m)}\right)$, where $\phi$ represents the parameters of the graph hypernetwork. Define a metric $d_{\mathcal{P}}$ on $\mathbb{R}^d$ by

$$d_{\mathcal{P}}(\phi, \phi') = \frac{1}{M} \mathbb{E}_{(x,y) \sim \mathcal{P}} \left| \sum_{m=1}^{M} \mathcal{L}_m(\phi) - \sum_{m=1}^{M} \mathcal{L}_m(\phi') \right|, \quad (12)$$

where $\mathcal{P} = \mathcal{P}_1 \times \cdots \times \mathcal{P}_M$ is the product probability measure, and $(x, y) = ((x^{(1)}, y^{(1)}), \ldots, (x^{(M)}, y^{(M)}))$. Define the covering number of a subset $E$ of $\mathbb{R}^d$ by closed ball of radius $\epsilon$ with respect to the metric $d_{\mathcal{P}}$ by

$$\mathcal{N}(\epsilon, E, d_{\mathcal{P}}) = \inf\{n : \exists \phi_1, \cdots, \phi_n, \forall \phi \in E, \exists j, d_{\mathcal{P}}(\phi, \phi_j) \leq \epsilon\} \quad (13)$$

and the capacity of $E \subset \mathbb{R}^d$ by

$$C(\epsilon, E) = \sup_{\mathcal{P}} \mathcal{N}(\epsilon, E, d_{\mathcal{P}}), \quad (14)$$

where the supremum is taken over all product probability measures on $(\mathcal{X}, \mathcal{Y})^M$. The capacity measures the complexity of the hypothesis space in much the same way as VC-dimension measures the complexity of a set of Boolean functions. Here our hypothesis space is indexed by $\phi \in \mathbb{R}^d$. Now we are ready to state the theorem from Baxter applied in our RECIPFL framework.

THEOREM A.1. *Let $D = \{D_m\}_{m=1}^{M}$ be generated by $N$ independent trials from $(\mathcal{X} \times \mathcal{Y})^M$ according to some product probability measure $\mathcal{P} = \mathcal{P}_1 \times \cdots \times \mathcal{P}_M$. If*

$$N \geq \max\left\{ \frac{4}{M\epsilon^2} \log \frac{4C\left(\frac{\epsilon}{4}, \mathbb{R}^d\right)}{\delta}, \frac{1}{\epsilon^2} \right\}, \quad (15)$$

*then*

$$\mathbb{P}\left( D : \sup_{\phi} |\mathcal{L}(\phi) - \hat{\mathcal{L}}(\phi, D)| > \epsilon \right) \leq \delta. \quad (16)$$

PROOF OF THEOREM 4.1. It suffices to bound $C\left(\frac{\epsilon}{4}, \mathbb{R}^d\right)$. Notice that by the Lipschitz assumption on the loss function $l$, $|\mathcal{L}_m(\phi) - \mathcal{L}_m(\phi')| \leq K\|\phi - \phi'\|$ for all $m = 1, \cdots, M$. This implies by (12), for all $\phi, \phi' \in \mathbb{R}^d$,

$$d_{\mathcal{P}}(\phi, \phi') \leq \frac{1}{M} \sum_{m=1}^{M} |\mathcal{L}_m(\phi) - \mathcal{L}_m(\phi')| \leq K\|\phi - \phi'\|. \quad (17)$$

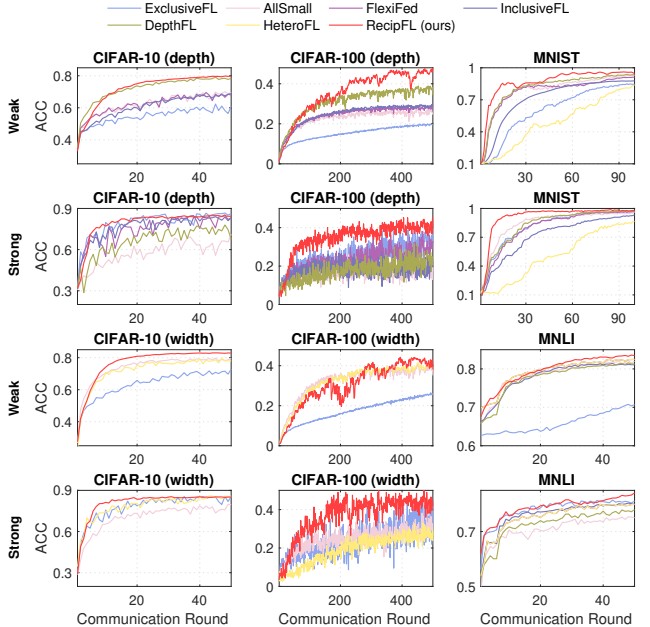

**Figure 7: Performance w.r.t. communication round**

So $\|\phi - \phi'\| \leq \frac{\epsilon}{K}$ implies $d_{\mathcal{P}}(\phi, \phi') \leq \epsilon$. Now take an integer $p > RK\sqrt{d}/\epsilon$ and decompose $[-R, R]^d$ as the union of $p^d$ congruent cubes by dividing $[-R, R]$ into $p$ pieces of equal length. The side length of these cubes is $2R/p$ and so each cube is contained in a ball of radius $R\sqrt{d}/p < \frac{\epsilon}{K}$ centered at the center of the cube. This proves the covering number $\mathcal{N}(\epsilon, E, d_{\mathcal{P}}) \leq \lceil RK\sqrt{d}/\epsilon \rceil^d$ for all $\mathcal{P}$. So, $C\left(\frac{\epsilon}{4}, \mathbb{R}^d\right) \leq \lceil 4RK\sqrt{d}/\epsilon \rceil^d$. $\square$

