# OpenReview forum: "How Few Davids Improve One Goliath: Federated Learning in Resource-Skew Edge Computing Environments"
_ACM.org/TheWebConf/2024/Conference — TheWebConf24 Oral_

### Official Review · Reviewer_YfGn · 2023-11-21

**Novelty:** 5
**Technical Quality:** 5

**Review:**

In this paper, the authors address the scenario of FL where a small number of powerful devices possess a substantial volume of data resources, while a large number of less capable devices are involved. The paper's primary contribution is the introduction of a novel framework called RecipFL, designed to efficiently derive weights for diverse client models using a graph hypernetwork. Additionally, the authors establish a generalization bound for RecipFL through theoretical analysis and validate its effectiveness through extensive experiments across a range of tasks.

Strengths:
(1) The paper highlights a compelling and practical scenario wherein a small number of powerful devices can support large models, while the majority of devices are constrained to smaller models.
(2) The authors introduce an innovative framework that is model-agnostic, facilitating collaboration among a wide range of neural network models.
(3) The experiments conducted in the study are comprehensive and meticulously executed, contributing to the paper's robustness.

Weaknesses:
(1) The paper would benefit from a more detailed explanation of how the hypernetwork can generate personalized weights for client models.
(2) While the paper provides insights into extreme cases (1 or 2 devices accompanied by a large number weak devices), it would be valuable to include a discussion of how the proposed framework operates in a wider range of practical cases to assess its generalizability and real-world applicability.

**Questions:**

(1) Is pre-training required for the graph hypernetwork to acquire the capability of generating personalized weights for client models? If so, could the authors elaborate on the specific training procedures? If not, how does the framework ensure that the weights generated by the hypernetwork effectively benefit the training of client models?
(2) Is the RecipFL framework limited to a single large model and a single small model? How does the framework accommodate a more general scenario where multiple model types and varying levels of computing power exist? For example, how well can RecipFL adapt to scenarios where a mix of models, such as ResNet-101, VGG-16, and LeNet-5, coexist within the system?
(3) The paper mentions that the server randomly samples clients per round. Is there a possibility that none of the strong devices are sampled in a round? If so, how does the framework address such cases?

**Reviewer Confidence:**

3: The reviewer is confident but not certain that the evaluation is correct

**Scope:**

4: The work is relevant to the Web and to the track, and is of broad interest to the community

---

### Official Review · Reviewer_k9GE · 2023-11-22

**Novelty:** 5
**Technical Quality:** 6

**Review:**

This paper manages to address the challenge of resource skewness in federated learning environments, where a small number of strong devices are accompanied by a large number of weak devices. This paper proposes RecipFL to facilitate reciprocal benefits between strong and weak devices in resource-skew scenarios which effectively generates weights for heterogeneous client models based on a graph hypernetwork, allowing weak devices to contribute meaningfully to the learning system and incentivizing strong devices to actively engage in federated learning.

**Pros:**
1. The problem of FL in resource-skew computing environments that the article endeavors to address is critical.
2. Adequate experimental analysis.
3. The code is open-source.

**Cons:**
1. The datasets are small and not practical.
2. Insufficient ablation study.

**Questions:**

1. I couldn't find enough motivation behind the design of the proposed graph hypernetwork in section 3.2, can you give more details?
2. The focus of RecipFL is on the contribution of weak devices to strong devices, but the experimental results show that the weak devices also improve significantly, why?

**Reviewer Confidence:**

2: The reviewer is willing to defend the evaluation, but it is likely that the reviewer did not understand parts of the paper

**Scope:**

3: The work is somewhat relevant to the Web and to the track, and is of narrow interest to a sub-community

---

### Official Review · Reviewer_PZ2U · 2023-11-22

**Novelty:** 6
**Technical Quality:** 5

**Review:**

This paper introduces a novel FL framework to address the resource-skew scenarios. Especially, the authors target an interesting problem of "Can strong devices benefit from weak devices in resource-skew computing environments?". The key technique is to allow clients to use heterogeneous models with a graph hypernetwork (GHN) to directly generate weights for clients.

Overall, the paper is well-motivated and tackling a novel, interesting and important problem in FL. I enjoyed reading it. The solution is novel and makes sense. The evaluation includes a sufficient number of baselines and is conducted both CV and NLP models, although the image datasets are mostly small.

**Questions:**

1. The explanation about the problem of ill-aligned layers (line 102-108) makes sense for depth-scaling heterogeneous-model FL approaches, but how about width-scaling approaches (in which the layers should be well aligned)?

2. Using heterogeneous models is not the only solution for resource heterogeneity issue in FL. How does it compare with other approaches (e.g. data sample selection and training pace controlling approaches)? Are they orthogonal?

Also, does it make sense to directly use a dynamic network architecture (e.g. [1](https://arxiv.org/abs/2102.04906), [2](https://dl.acm.org/doi/10.1145/3570361.3592529), [3](https://dl.acm.org/doi/abs/10.1145/3447993.3483249)) that can automatically adapt their sizes on different devices in FL? What are the disadvantages of this approach as compared with yours?

3. Would the model weight generation process of GHN introduce high overhead? Especially when the number of clients increases? Meanwhile, showing the training overhead of this model will be better.

4. Why the results of FlexiFed and HeteroFL on MNLI are exactly the same in Table 3?

**Reviewer Confidence:**

3: The reviewer is confident but not certain that the evaluation is correct

**Scope:**

3: The work is somewhat relevant to the Web and to the track, and is of narrow interest to a sub-community

---

### Official Review · Reviewer_LkLc · 2023-11-23

**Novelty:** 4
**Technical Quality:** 5

**Review:**

The paper introduces RecipFL, a federated learning framework designed to manage resource-skewed environments where a few strong devices (like enterprise-level data centers) coexist with numerous weak devices (like mobile and Web-of-Things devices). RecipFL employs a server-side graph hypernetwork to generate weights for heterogeneous client models, accommodating arbitrary model scaling strategies. This framework addresses the challenge of aligning and aggregating models of varying capacities and architectures, enabling knowledge transfer from large to small models through regularization and distillation techniques. Theoretical analysis establishes a generalization bound for RecipFL, and extensive experiments demonstrate its efficacy across various datasets for tasks like image classification and natural language inference.

Strengths:

RecipFL's approach to managing skewed resources in federated learning is novel and addresses a significant gap in existing methods.

The flexibility to work with different model sizes and architectures enhances the applicability of RecipFL in diverse computing environments.

The use of cross-entropy loss and KL-divergence loss for knowledge transfer from large to small models is an effective strategy to improve the learning of weak devices.

Establishing a generalization bound for the RecipFL method provides a solid theoretical basis for its application.

Weaknesses:
Performance on Large Device Graphs: Given that graph neural networks (GNNs) are prone to overfitting on large graphs, how does RecipFL perform on large device graphs?

Complexity in Graph Hypernetwork Implementation: The implementation of a graph hypernetwork in RecipFL is innovative, but it introduces significant complexity. This complexity could impact the framework's usability, making it challenging for practitioners to implement and scale in varied federated learning environments.

Real-World Generalizability and Versatility: The experiments conducted may not fully capture the framework's performance in real-world heterogeneous environments. It's crucial to understand how RecipFL adapts to environments with different network conditions and a broader spectrum of device capabilities.

Dependence on Server-Side Processing Capacity: RecipFL relies heavily on server-side computations, which could be a limiting factor in scenarios where server capabilities are constrained. This dependence may hinder the framework's applicability in edge computing environments with limited server resources.

Resource Demands on Weak Devices: Although RecipFL is designed to enhance the learning of small models on weak devices, the process itself, including knowledge transfer and distillation techniques, might still be resource-intensive. This aspect could be a significant barrier, especially for devices with very limited computational power or storage.

Efficiency in Knowledge Transfer Techniques: The efficiency of the employed knowledge transfer techniques from large to small models, especially in terms of computational overhead and the impact on the learning of weak devices, requires further exploration to ensure it does not disproportionately burden weak devices.

**Questions:**

Performance on Large Device Graphs:

Given that graph neural networks (GNNs) are prone to overfitting on large graphs, how does RecipFL perform on large device graphs?
Are there any built-in mechanisms or strategies in RecipFL to prevent overfitting when dealing with large-scale device graphs?

Complexity in Graph Hypernetwork Implementation:

Could the authors elaborate on how practitioners can overcome the complexity in implementing the graph hypernetwork, especially in diverse federated learning environments?
Is there a way to simplify the graph hypernetwork or provide more user-friendly tools or documentation to aid its implementation?

Real-World Generalizability and Versatility:

How does the RecipFL framework adapt to real-world heterogeneous environments with varying network conditions and device capabilities?
Are there plans to conduct more extensive real-world tests to validate the framework's versatility and generalizability?

Dependence on Server-Side Processing Capacity:

Given the heavy reliance on server-side computations, how does RecipFL plan to address scenarios with limited server capabilities, especially in edge computing environments?
Could the framework be adapted to distribute more computational tasks to client devices without significantly impacting their performance?

Resource Demands on Weak Devices:

How does the framework manage the resource demands on weak devices, particularly in terms of computational overhead and storage requirements?
Are there optimizations within RecipFL that specifically target resource conservation on weaker devices?

Efficiency in Knowledge Transfer Techniques:

Could the authors detail the measures taken to ensure that the knowledge transfer techniques are efficient and don't overly burden weak devices?
Are there any specific strategies employed to optimize the computational overhead during knowledge transfer?

**Reviewer Confidence:**

4: The reviewer is certain that the evaluation is correct and very familiar with the relevant literature

**Scope:**

3: The work is somewhat relevant to the Web and to the track, and is of narrow interest to a sub-community

---

### Official Review · Reviewer_Ytf9 · 2023-11-25

**Novelty:** 4
**Technical Quality:** 5

**Review:**

Thank you for your submission.

This work addresses the challenge of imbalance contributions and capabilities in federated learning with a proposal named RECIPFL that uses a server side graph hypernetwork to capture the structure of the models and is agnostic to model scaling.

Main comments:

- Number of strong devices. The authors have 1 or 2  strong devices in their modelling and evaluation but there is no clear justification why this is the only scenario to be addressed. I would expect a wider evaluation with different distributions of strong/weak devices and/or a clear justification why studying the very specific scenario of 1-2 devices is sufficient.

- Capacity of weak devices. The weak devices are assumed to be homogeneous in terms of data and capabilities. This does not seem a realistic scenario and at the least a justification of why this is sufficient is needed.

- Relationship to the web. The problem tackle by this paper is very relevant to the web, but the paper makes limited connections in both its explanations and evaluation. The paper seems oriented to a ML audience and leaves concepts that might not be equally familiar in a web-related conference (eg, hot/dense vector).


- Evaluation datasets. There is barely any information about the datasets used for the evaluation and almost no discussion of their adequacy. A reference and the topic of the dataset is an insufficient description.


Other comments:
- References missing for the claims that existing works assume that weak and strong devices are equally distributed in section 2.2.

**Questions:**

- Why is analyzing scenarios with 1 or 2 devices sufficient?
- Why assuming homogeneous weak devices is a reasonable assumption?

**Reviewer Confidence:**

1: The reviewer's evaluation is an educated guess

**Scope:**

2: The connection to the Web is incidental, e.g., use of Web data or API

---

### Decision · Program_Chairs · 2024-01-22

**Decision:**

Accept (Oral)

**Comment:**

The paper addresses a federated RL framework in the edge environment, where it showcases that a combination of a few strong devices and many weak devices can effectively support large models.

 The reviewers appreciated the important and compelling problem, model-agnostic properties, and comprehensive evaluations. In the meantime, multiple reviewers commonly pointed out the specific combination of strong + weak devices, calling for justifications and generalizability toward more various real-world scenarios. evaluations that are limited to simulated environments, leaving various real-world factors and dynamicity uncovered. I also greatly appreciate the authors' effort in thoroughly communicating with the reviewers.

 Overall, given the ratings, review comments, and discussion, I believe that this paper's merits somewhat outweigh its concerns. My AC recommendation is set accordingly.